# Metasurface-based dual-sense circularly polarized antenna for MIMO/full-duplex applications

**Duc-Nguyen Tran-Viet**[1], **Hong Nguyen Tuan**[2], **Dinh Nguyen Quoc**[3], **Dat Nguyen Tien**[4], **Hung Tran-Huy**[4]*

1 Faculty of Radio Electronics Engineering, Le Quy Don Technical University, Ha Noi, Viet Nam, 2 Center for High Technology Development, Vietnam Academy of Science and Technology (VAST), Hanoi, Vietnam, 3 Advanced Wireless Communications Group, Le Quy Don Technical University, Ha Noi, Viet Nam, 4 Faculty of Electrical and Electronic Engineering, PHENIKAA University, Yen Nghia, Ha Dong, Hanoi, Vietnam

* hung.tranhuy@phenikaa-uni.edu.vn

**Data Availability Statement:** All relevant data are within the paper.

## Abstract

This paper introduces a two-element antenna array with dual-sense circular polarization, wideband operation, and high isolation characteristics. The antenna consists of two conventional truncated corner patches and an extra layer of metasurface (MS) located above the radiating patches. The overall dimensions of the proposed antenna are $0.92 \lambda_0 \times 0.73 \lambda_0 \times 0.05 \lambda_0$ and the element spacings are $0.02 \lambda_0$ and $0.39 \lambda_0$ with respect to edge-to-edge and center-to-center spacings. For validation, measurements on a fabricated antenna prototype are carried out. The measured data demonstrate that the presented MS-based antenna has a wide operating bandwidth of 14.5% with high isolation of better than 26 dB. The excellent performance could be concluded from the results of the investigation, which indicates that the proposed MS-based antenna could be a good candidate for multiple-input multiple-output (MIMO) and full-duplex applications.

## Introduction

In the last few years, full-duplex and multiple-input multiple-output (MIMO) technologies have been considered one of the most essential techniques for their ability to improve channel capacity in modern wireless communications [1–3]. Among the released studies on these types of antennas, microstrip patch antennas are the most investigated and discussed configurations [4, 5]. Besides the benefits of microstrip patch MIMO antennas, two common challenges still attract lots of attention from antenna researchers worldwide. First, while allocating the radiating elements closer to each other to achieve the overall size reduction, there would be a strong mutual coupling between the neighboring patches, significantly reducing antenna performance. Consequently, the mutual coupling effect must be alleviated. The remaining challenge is to employ circularly polarized waves to make use of the advantages in both line-of-sight and multipath propagations. Hence, this paper is motivated by the idea of proposing a printed antenna array that can produce CP diversity and low mutual coupling within a wide operating frequency band.

**Funding:** This work was supported by the Vietnam Academy of Science and Technology (VAST) under grant number TANQP.02/23–25.

**Competing interests:** The authors have declared that no competing interests exist.

It has been reported in the literature that there are several methods to mitigate the mutual coupling effect between the radiating elements, which can be categorized into two major groups. In the first classified solution of coupling reduction, there is no need for circuit modification to be added to the antennas. Those solutions either occupy different modes of electromagnetic waves [6] or change the orientation of the radiating elements [7–9] to improve the isolation of the whole system. The self-decoupled methods offer certain advantages of low complexity in antenna design and a notable coupling reduction. By contrast, they also lead to an increase in the overall dimensions, which may need to be revised for compact devices. The remaining category requires extra decoupling networks. Commonly, those additional structures are located within the same layer as the radiators, including defected ground structure (DGS) [10–12], parasitic elements [13–17], and neutralization lines [18–20]. The primary operating principles of those configurations are to counteract or block the interaction between the array elements in terms of surface current within the dielectric substrate, eventually bringing a significant improvement in isolation. Recently, metasurface (MS) has been widely used by antenna designer as its superior features [21, 22]. The decoupling MS can be positioned in the same layer [23, 24] or above the patches [25, 26]. Despite the recent great efforts, the low mutual coupling structures in the reported studies are still restricted in narrow bandwidth (BW). In addition, other drawbacks of those designs are sophisticated decoupling networks and linear polarization operations.

It is noticeable that antennas that produce CP operation have been investigated more and more in the last few years. Unlike linearly polarized (LP) antenna designs, it is challenging to apply self-decoupling methods to CP antennas [27], which is why decoupling networks are the most appropriate approach in CP microstrip patch configurations. The use of DGS structures has been implemented in [28, 29]. However, those designs can operate in narrow BWs of less than 2%. Although introducing either parasitic elements [30] or MS within the same layer with the radiating elements [31, 32] is an effective solution to improve operating BW, large overall dimensions could be their vital drawbacks. Another approach that offers wide BW is to use a metallic post [33], but this structure yields an antenna with a very high profile.

This paper presents a two-element dual-sense CP antenna with significantly high isolation in a wide operating band ranging from 5.1 to 5.9 GHz. The main radiating elements are composed of two conventional truncated corner microstrip patches, in which the cutting positions are different to ensure both left-hand CP (LHCP) and right-hand CP (RHCP) radiations. For mutual coupling reduction and BW enhancement, a pair of $2 \times 4$ unit cell MS structures are positioned above the radiators on the top layer of the second substrate. By properly designing the MS layer, the proposed design have several advantages in comparison with the related works. Here, wideband operation with high isolation can be achieved while keeping small element spacing. The antenna is first simulated using full-wave High-Frequency Structure Simulator (HFSS) and then verified by measurements.

## Single-element design

It is well-known that a single patch antenna suffers from poor performance in terms of both impedance and AR BWs. One of the most effective ways to achieve wideband performance is to combine the patch with the MS [34, 35]. Fig 1 shows three different antennas, which are designated as Ant-1, -2, and -3. These antennas are designed on the Taconic-RF35 substrates, which have a thickness of 1.52 mm, a dielectric constant of 3.5, and a loss tangent of 0.0018. The optimized dimensions are summarized in Table 1.

The performances in terms of reflection coefficient and AR of Ant-1, -2, and -3 are shown in Fig 2. The simulated results are achieved from the full-wave simulator HFSS. The AR is

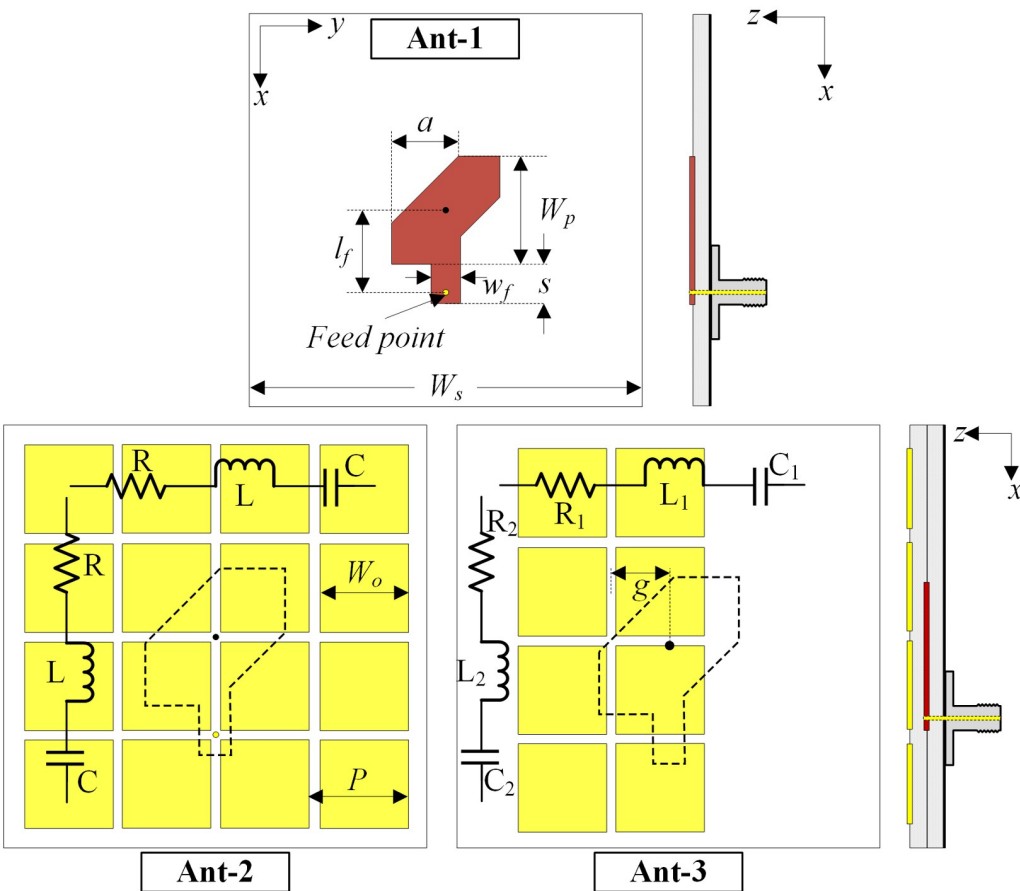

**Fig 1. Geometry of different single-element CP antennas.**

calculated in the forward direction with *Phi* = *Theta* = 0 deg. For Ant-1, the radiating element is a square patch with truncated corners to radiate CP waves. This design has a narrow operating BW of less than 5%. With the presence of MS positioned above the patch, significant performance improvement can be achieved. The wideband principle is based on the combination of two adjacent bands, which are the lower band of the primary radiating patch and the higher band of the MS. Thorough investigation of the wideband mechanism has been mentioned in

**Table 1. Optimized dimensions of different single-element CP antennas (unit: mm).**

| Parameter | Ant-1 | Ant-2 | Ant-3 |
|---|---|---|---|
| Ws | 40 | 32 | 40 |
| Wp | 12.8 | 13 | 13 |
| a | 7.2 | 7 | 7 |
| lf | 8.4 | 8.5 | 8.5 |
| wf | 3 | 3 | 2.8 |
| s | 3.8 | 4 | 4.5 |
| P | | 8 | 8 |
| Wo | | 7.5 | 7.5 |
| g | | | 5.3 |

(a)

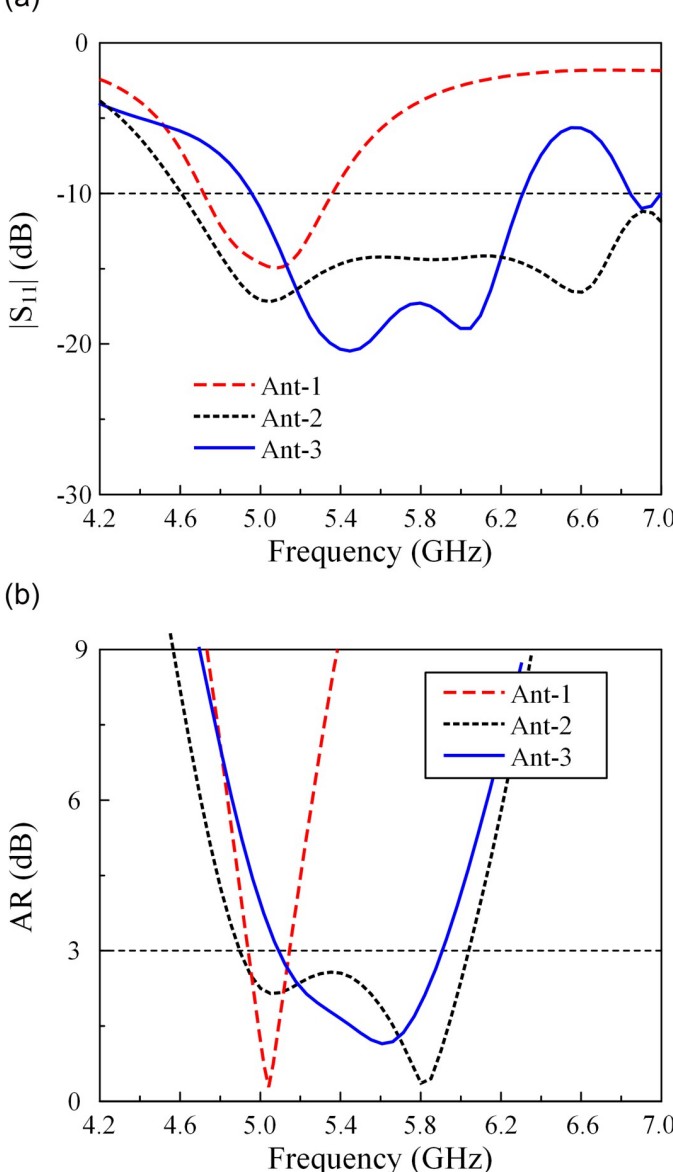

(b)

**Fig 2. Simulated performance of different single-element CP antennas.** (a) Reflection coefficient ($|S_{11}|$) and (b) Axial-ratio (AR).

[34, 35]. Both Ant-2 and Ant-3 employ the square unit cell. In the case of Ant-2, a 4 × 4 unit-cell MS is centered within the patch. In contrast, Ant-3 utilizes a 2 × 4 unit-cell MS, which is positioned off-center within the patch. It is worth noting that the configuration and position of the MS are critical to the compactness of the two-element antenna, which will be discussed in the following Section.

The operating mechanism of the MS-based antenna has been thoroughly investigated in [34]. The radiating patch with corner truncation works in the fundamental $TM_{11}$ mode for CP radiation. The MS acts as a parasitic element, which is coupled with the CP source to produce a higher operational band. The equivalent circuits of the MSs in Ant-2 and -3 are presented in

Fig 1. For Ant-2, the MS consists of $4 \times 4$ unit cells with similar equivalent circuits in horizontal and vertical directions are similar. Thus, it is positioned at the center of the patch and coupled with orthogonal modes $TM_{01}$ and $TM_{10}$ of the primary radiating patch to produce higher CP band operation. For Ant-3, the MS is asymmetric with $4 \times 2$ unit cells, leading to the difference in the equivalent circuits in horizontal and vertical directions. Therefore, it is positioned off-centered with the primary radiating patch, $g$. In this case, the coupled fields in the MS can be controlled by tuning $g$ until they are equal in magnitude and 90° out of phase for CP radiation.

## Two-element antenna design

It has been discussed in the previous Section that both Ant-2 and -3 exhibit wideband performance with different configurations and positions of the MSs. Next, they are used to design two-element antennas with dual-sense CP, as illustrated in Fig 3. This Section will demonstrate that the configuration and position of the MS are very important in designing a

(a)

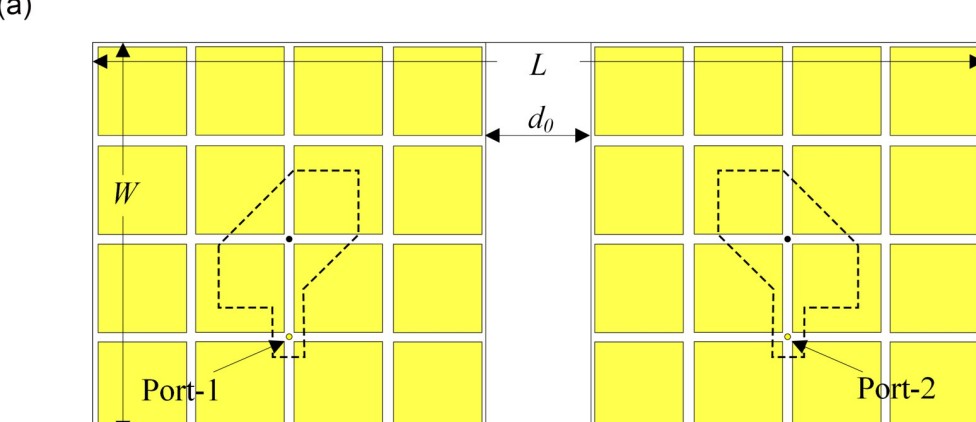

(b)

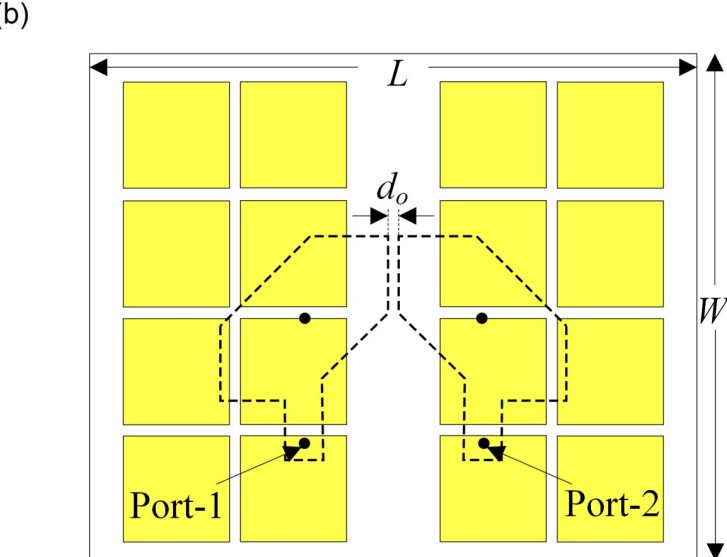

**Fig 3. Configurations of different two-element antennas.** (a) Ant-4 and (b) Ant-5.

two-element antenna, which exhibits high isolation with small element spacing. As the designs aim for MIMO and full duplex applications, high isolation is required. Thus, Ant-4 and Ant-5 are optimized so that their isolations are always better than 25 dB across the operating BW.

For dual-sense antennas, the radiating elements are two conventional microstrip square patches, which are truncated in different sets of corners to radiate LHCP and RHCP waves, respectively. The antenna is fed by two 50-$\Omega$ SMA conductors, in which the LHCP patch is excited by Port-1 and Port-2 excites the RHCP patch. The optimized dimensions of Ant-4 are $L$ = 70, $W$ = 32, $W_p$ = 12.2, $l_f$ = 9.6, $s$ = 4.0, $w_f$ = 2.6, $a$ = 7.6, $d_0$ = 6.5, $P$ = 8, $W_0$ = 7.5 (unit: mm). The optimized dimensions of Ant-5 are $L$ = 50, $W$ = 40, $W_p$ = 13.1, $l_f$ = 8.5, $s$ = 4.5, $w_f$ = 2.8, $d_0$ = 1, $a$ = 6.8, $P$ = 8.7, $W_0$ = 8.2, $d_0$ = 14, $g$ = 5.1 (unit: mm).

The simulated performances in terms of reflection coefficient, transmission coefficient, and AR of Ant-4 and -5 are presented in Fig 4. The data indicate that both antennas have similar operating BW, which is defined by the overlap between -10 dB and 3-dB AR BWs. Here, the operating BWs of Ant-4 and -5 are quite similar. Regarding the isolation, both designs have isolation of better than 25 dB across the operating BW. However, one of the most remarkable things is that the element spacings are significantly different for these designs despite having similar isolation. Ant-4 has an edge-to-edge spacing of 0.12 $\lambda_0$ and a center-to-center spacing of 0.71 $\lambda_0$. Meanwhile, the figures for Ant-5 are considerably smaller of 0.02 $\lambda_0$ and 0.39 $\lambda_0$ for edge and center spacings, respectively. As a result, the overall dimensions of Ant-5 are smaller than those of Ant-4.

## Antenna operation characteristic

To demonstrate the effectiveness of using MS in increasing the operating BW and isolation of the patch antenna, Fig 5 shows the performance of the two-element patch antenna as the reference antenna and the two-element MS-based antenna (Ant-5). Note that these antennas are optimized with a similar edge-to-edge element spacing. The operating BW is significantly increased from 7.2% (5.28–5.68 GHz) for the patch antenna to 16.2% (5.1–6.0 GHz) for Ant-5. Concerning isolation, the antenna with MS shows much better isolation than the other. Within the operating BW, the isolation of the reference antenna is about 8.5 dB. Meanwhile, Ant-5 has an isolation of greater than 25 dB across the operating BW from 5.1 to 6.0 GHz.

The isolation enhancement can be explained by observing the current distribution on the reference antenna and Ant-5. Fig 6 illustrates the simulated current vectors J-surf of these antennas at 5.5 GHz when Port-1 is excited. Note that with Port-1 excitation, the radiated wave is LHCP, while the remaining port produces RHCP radiation. As seen in Fig 6(a), the reference antenna observes a strong coupling current on the non-excited patch at different phases of 0˚ and 90˚. Thus, this antenna suffers from extremely high mutual coupling. On the other hand, Ant-5 with MS has different current distributions. Here, the electromagnetic field from the excited element tends to couple with the MS rather than the non-excited element. Thus, the isolation for Ant-5 is significantly improved. Besides, looking at the surface current distribution in Fig 6, the rotation direction of the vector current on the patch and the MS is clockwise when the phase switches from 0˚ to 90˚, which proves the LHCP characteristic of the antenna toward the +$z$-direction.

## Optimization process

### Matching optimization

For the final realization of the proposed antenna, the parametric study stage has been implemented on different dimensions. In Fig 7, the simulation results with different values of the

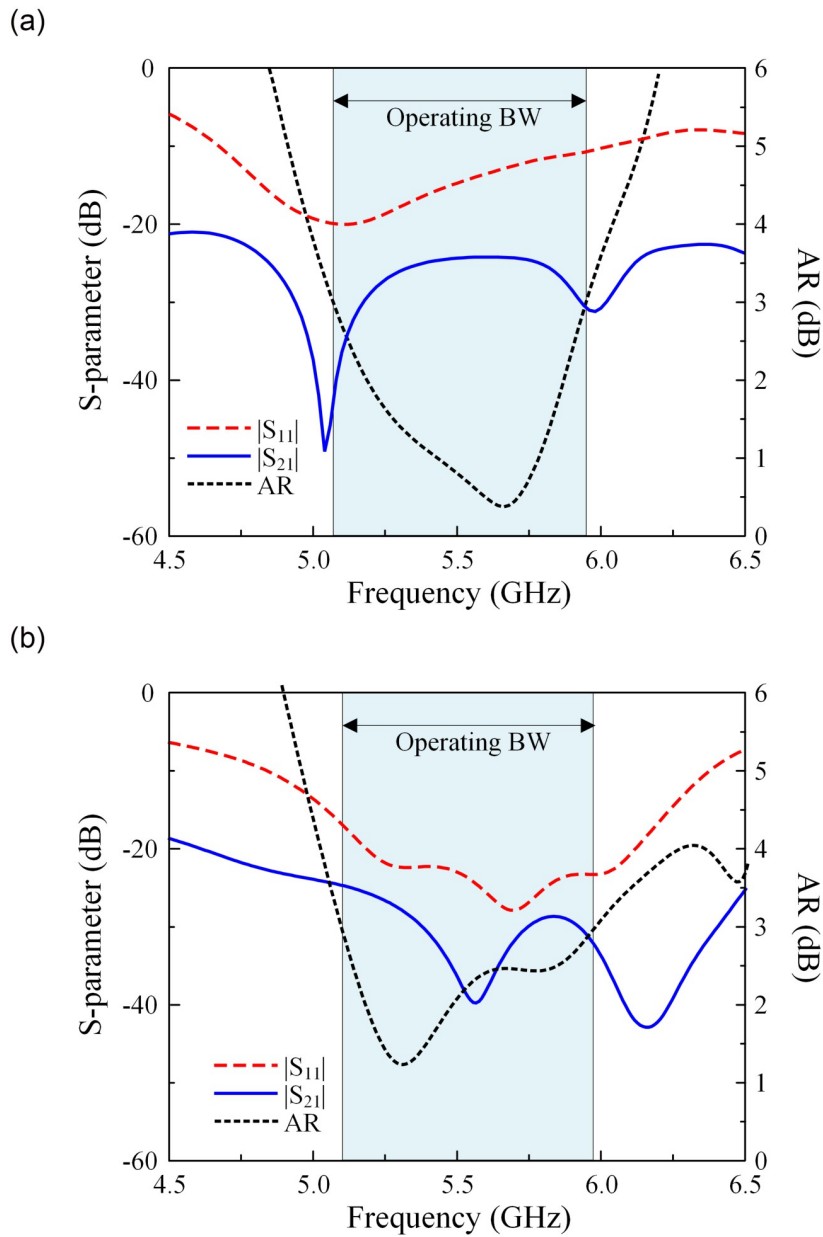

**Fig 4. Simulated performance of (a) Ant-4 and (b) Ant-5.**

stub length $l_s$ and feeding position $l_f$ are demonstrated. Here, changing the stub length and the feeding position will directly affect the antenna's input impedance. Thus, the reflection coefficient can be adjusted. The results for transmission coefficients are always lower than -20 dB within the investigated frequencies, which is why the figures for $|S_{21}|$ are not shown for the sake of brevity. Meanwhile, changing these parameters results in different antenna input impedances. As illustrated, those variants in the stub length $l_s$ and feeding position $l_f$ have a significant effect on the matching performance. With the increase of $l_s$ to 5.1 mm, the overall $|S_{11}|$ performance of the antenna is much better. Regarding the feeding position, the best performance in terms of impedance BW could be obtained when $l_f$ is set at 8.5 mm.

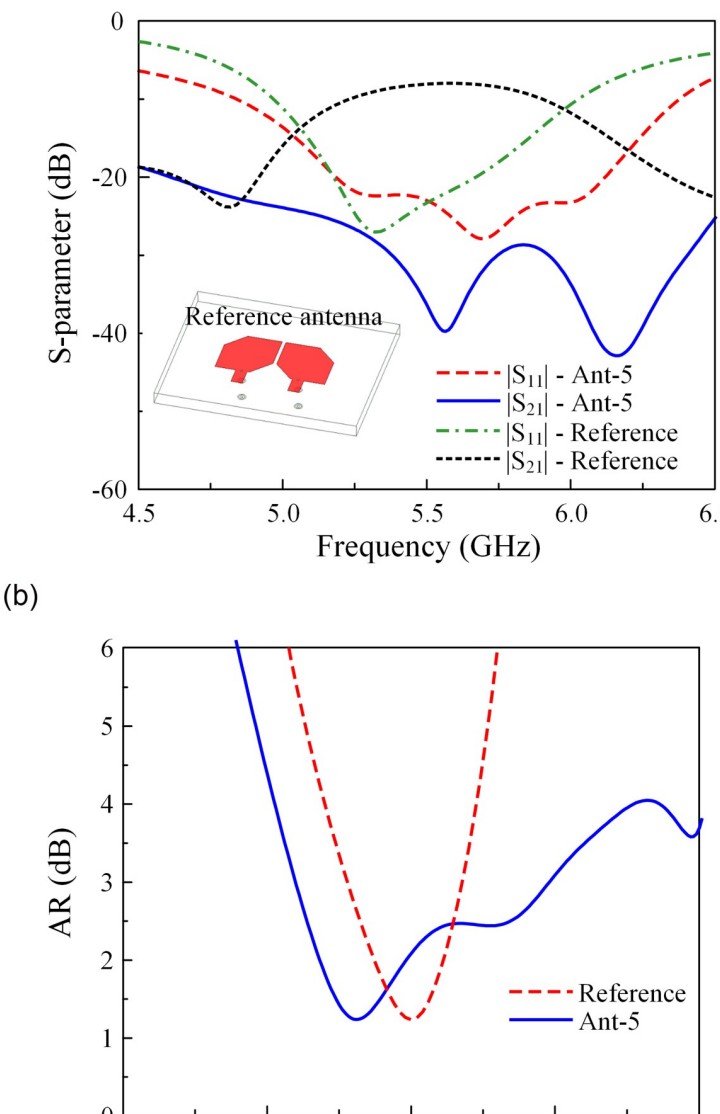

**Fig 5. Simulated (a) S-parameter and (b) AR of the reference antenna and Ant-5.**

## CP optimization

Another observation to determine the CP performance of the proposed antenna is the dimension of each unit cell of the MS. It is noted that the wide AR BW is achieved by producing two adjacent resonances, which are respectively produced by the radiating patch and the MS. For demonstration, the simulated AR results of the antenna with different values of $W_0$ are shown in Fig 8. As observed, the variation of $W_0$ has significant effect on the higher operating frequency, which shifts towards the lower band when increasing $W_0$. The best 3-dB AR operating band can be satisfied when $W_0$ is set to 8.2 mm.

(a)

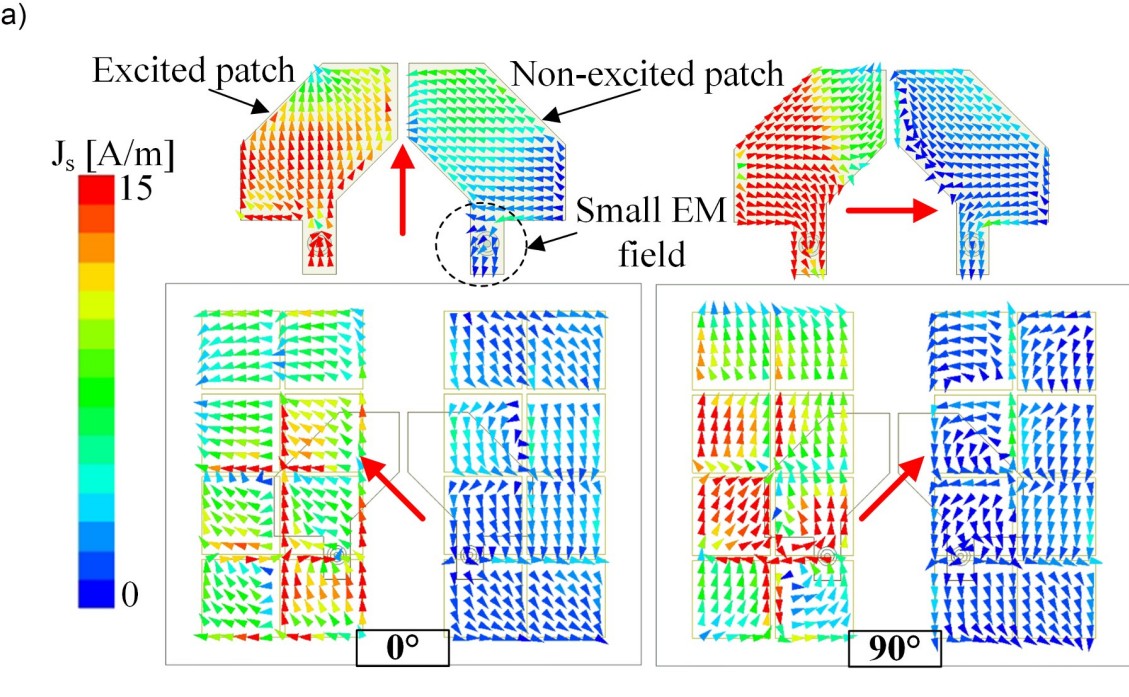

(b)

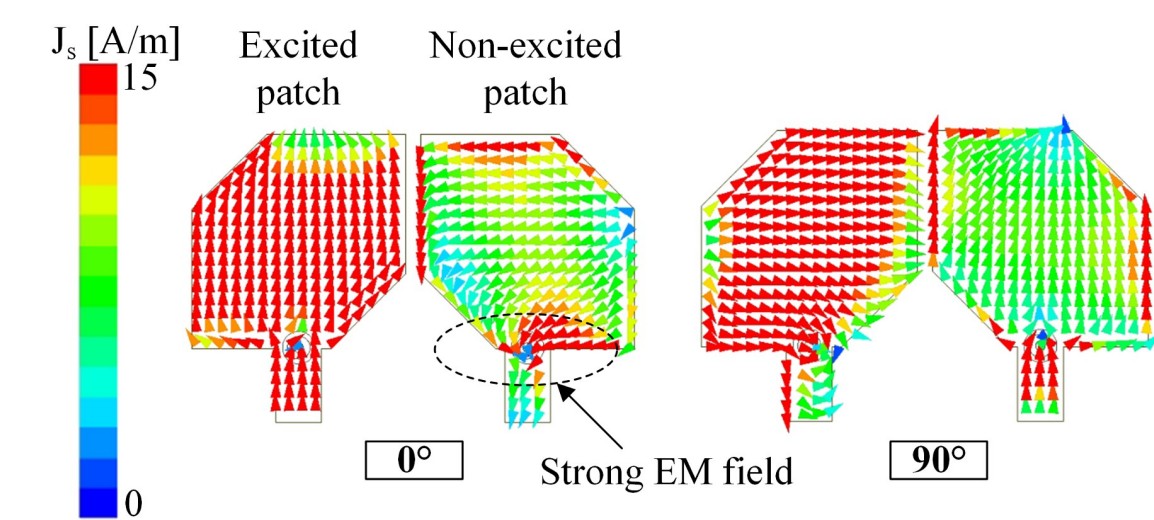

**Fig 6. Simulated surface current distribution at 5.5 GHz with Port-1 excitation.** (a) Ant-5 (b) reference antenna.

## Isolation optimization

In fact, the isolation can be improved when the distance between the elements is increased. However, the antenna size will eventually be increased. As discussed in the previous section, the mutual coupling can be reduced by introducing the MS layer. The electromagnetic field will couple to the MS rather than the non-excited element. Thus, high isolation can be achieved. Further investigation on the effect of the number of unit cells on the isolation is

(a)

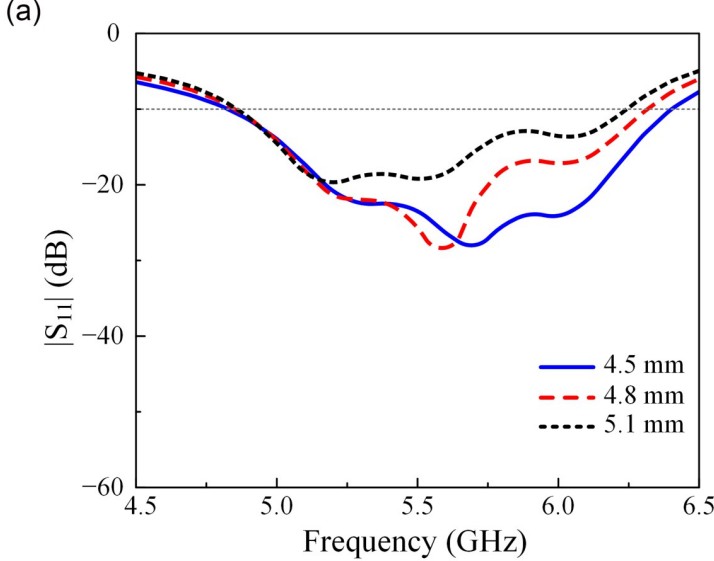

(b)

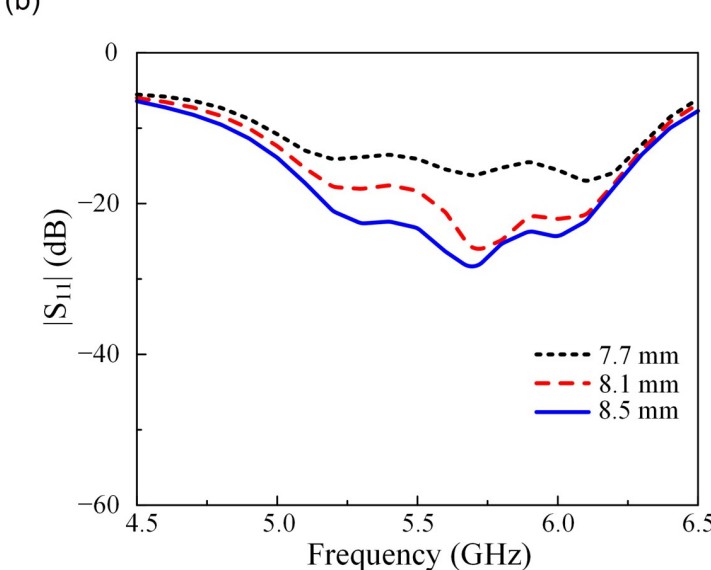

Fig 7. Simulated $|S_{11}|$ results of Ant-5 with different lengths of (a) $l_s$ and (b) $l_f$.

studied, and the results are presented in Fig 9. Here, three different MS structures are utilized with $2 \times 2$, $2 \times 3$, and $2 \times 4$ unit-cells. As observed, the greater the number of unit cells used, the better isolation can be attained.

The isolation improvement can be quantitatively explained based on the current distribution of the MS. Fig 10 shows the simulated current distributions at 5.5 GHz for Ant-5 with different types of MS. In all cases, the current is highly concentrated on the MS layer. If a greater number of unit cells are employed, more power will be coupled to the MS, leading to higher isolation. In comparison with the $2 \times 4$ MS, the antenna with $2 \times 5$ MS exhibits slightly better isolation. Along with that, the antenna size will be increased. Consequently, the $2 \times 4$ MS is chosen as the optimal design.

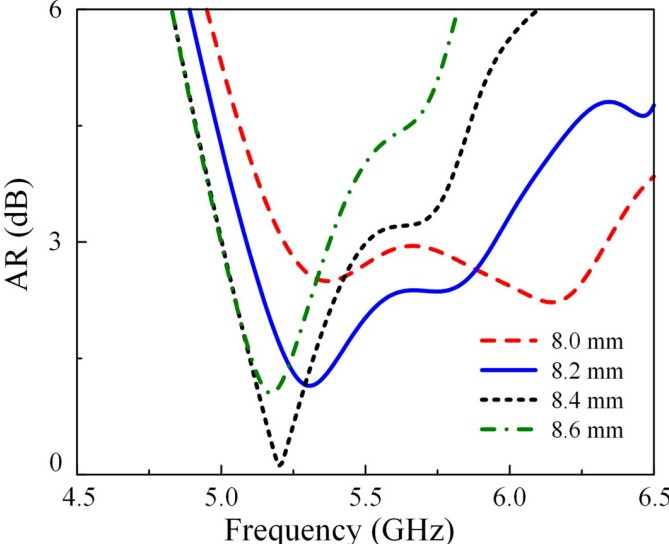

**Fig 8. Simulated AR results of Ant-5 with different values of $W_0$.**

## Measured results

To validate the antenna properties, a prototype is fabricated and measured. The photographs of the fabricated antenna are shown in Fig 11. The comparison between simulations and measurements indicates that the results are almost identical, with a small difference. The reason behind this difference is attributed to the tolerances in fabrication and measurement setups.

## S-parameter and far-field results

The results of simulated and measured S-parameter, AR, and realized gain are illustrated in Fig 12. During the far-field measurement, only one port is excited and the other one is

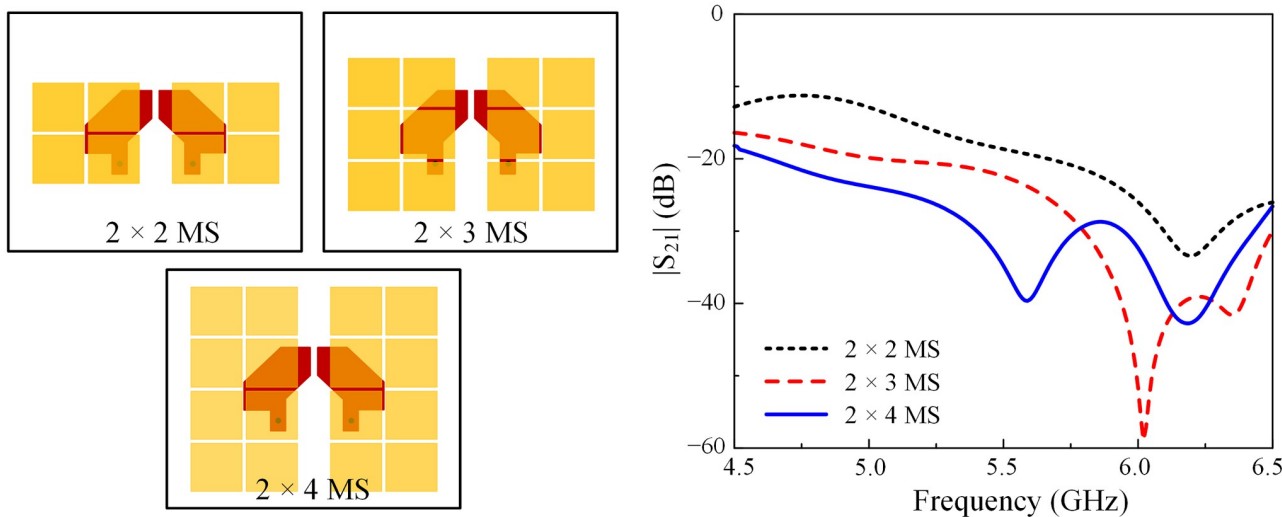

**Fig 9. Simulated transmission coefficients of Ant-5 with different numbers of unit cells.**

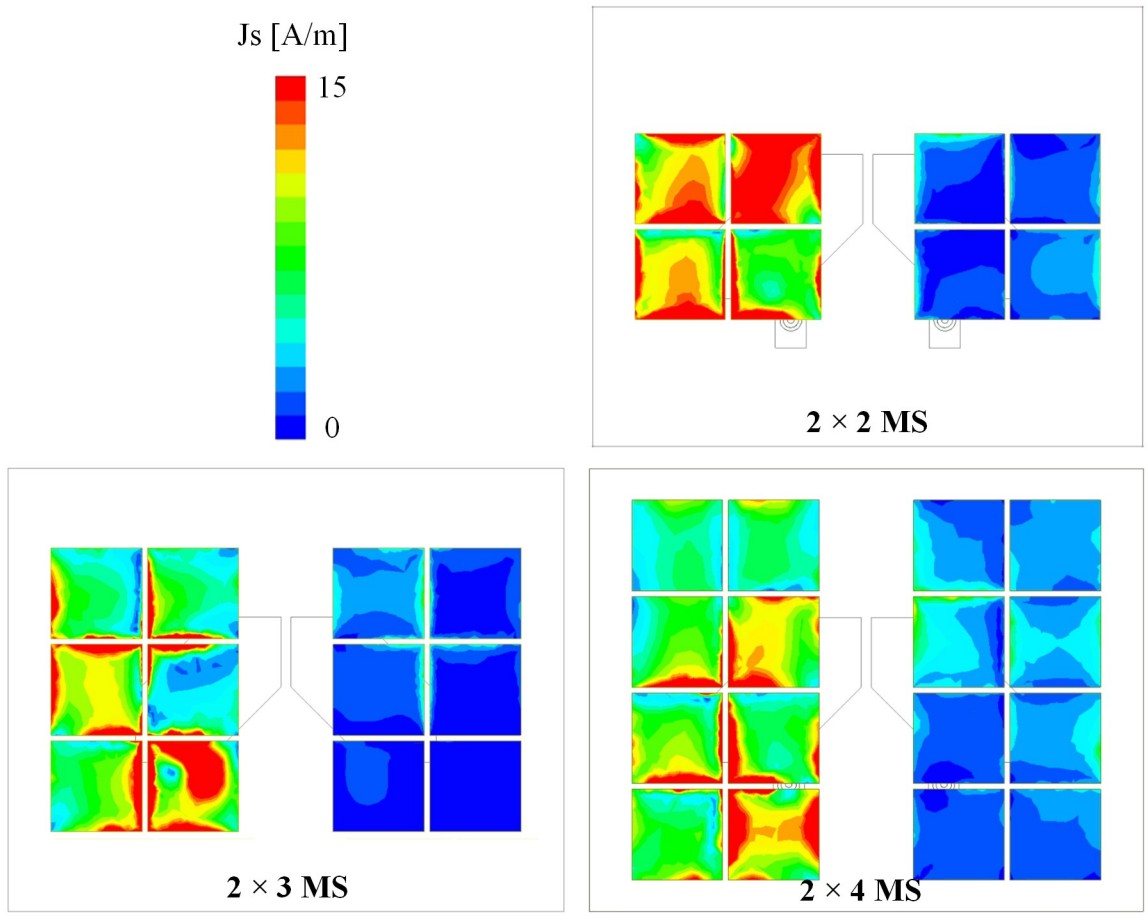

**Fig 10. Simulated current distributions at 5.5 GHz for Ant-5 with different types of MS.**

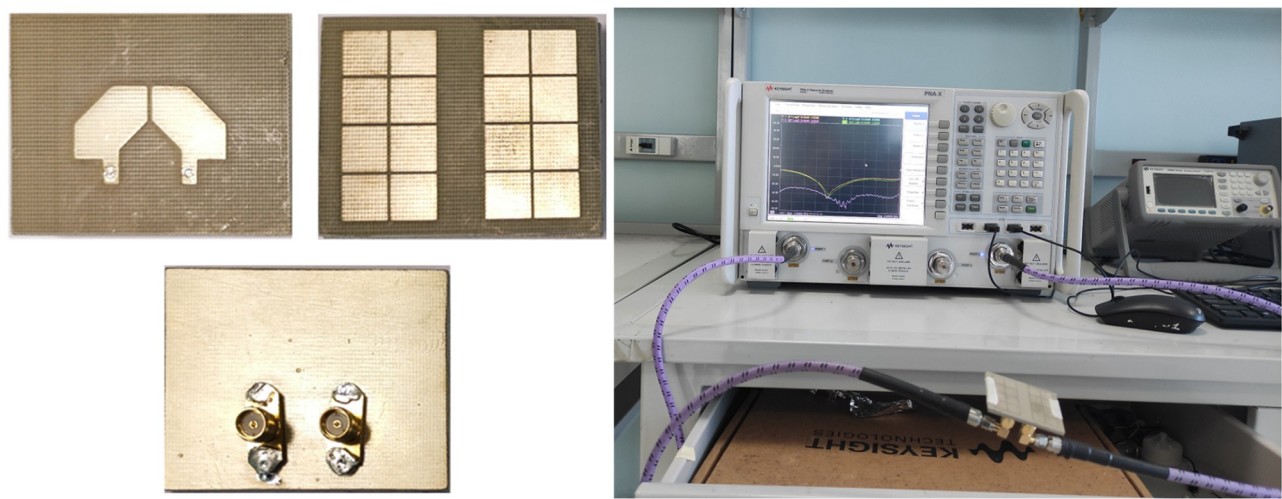

**Fig 11. Photographs of the fabricated MS-based MIMO CP antenna.**

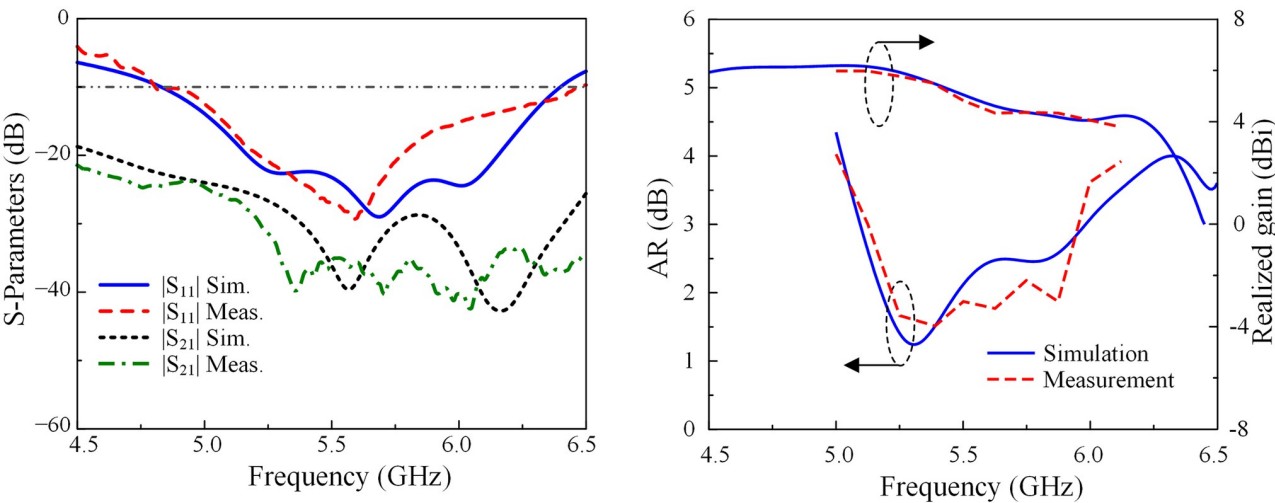

**Fig 12. Simulated and measured S-parameters, axial ratio, and gain of the proposed antenna.**

terminated with a 50-Ω load. The measured results demonstrate that the antenna has an impedance BW of 28.6%, ranging from 4.8 to 6.4 GHz. The isolations within this band are consistently higher than 23 dB. It can be concluded from the measured results that the CP operating BW is 14.5%, ranging from 5.1 to 5.9 GHz. Within this band, the isolation is better than 26 dB with a maximum value of 40 dB. Besides, the realized gain is also higher than 4.8 dBi. The peak realized gain within the AR BW is 6.0 dBi at 5.1 GHz, while the overall gains in the whole CP band are always higher than 4.2 dBi.

The gain radiation patterns in two principal planes of $x-z$ and $y-z$ at 5.2 GHz for Port-1 excitation are plotted in Fig 13. An observation of the case when Port-1 is excited shows that

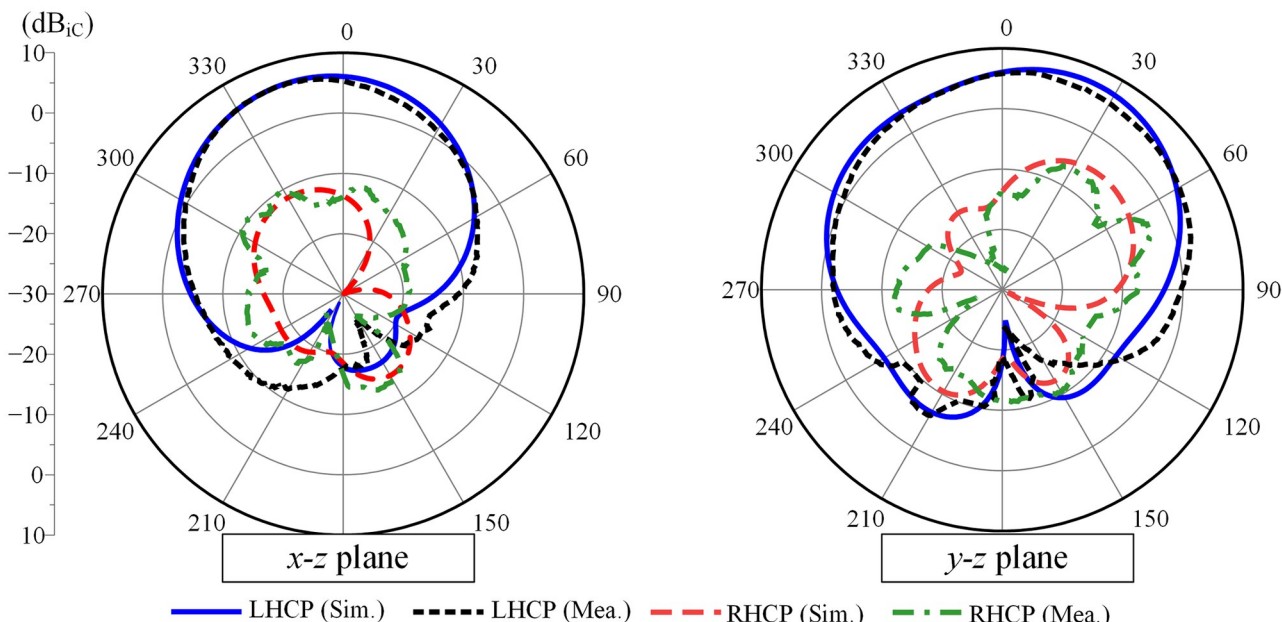

**Fig 13. Simulated and measured radiation patterns of the proposed antenna at 5.2 GHz.**

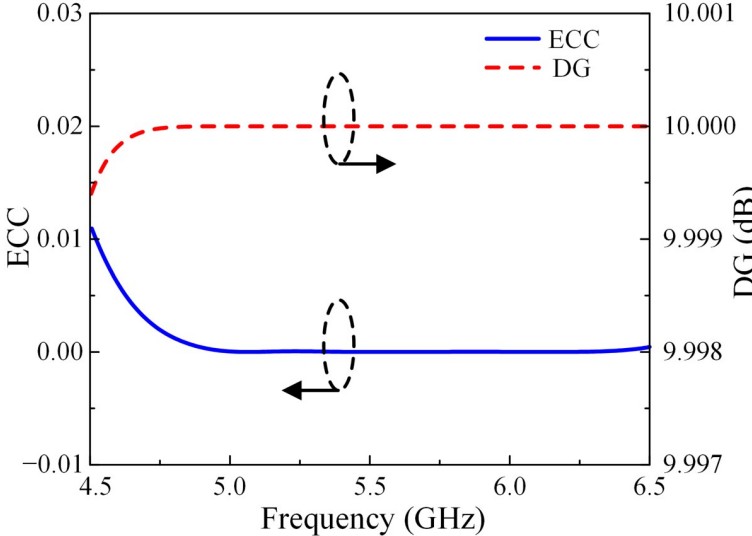

**Fig 14. ECC and DG of the proposed antenna.**

LHCP is the dominant radiation in the broadside direction. The polarization discrimination in the broadside direction is 16 dB, and the front-to-back ratio is better than 18 dB.

## MIMO parameters

To evaluate the feasibility to operate in MIMO systems of the proposed antenna, the MIMO parameters, including envelope correlation coefficient (ECC) and diversity gain (DG), are considered as calculated in Eqs (1) and (2) [36].

$$\rho_{ij} = \frac{\left|R_{ii}^* * T_{ij} + T_{ji}^* * S_{jj}\right|^2}{(1 - \left|R_{ii}\right|^2 - \left|T_{ji}\right|^2)(1 - \left|R_{jj}\right|^2 - \left|T_{ij}\right|^2)} \tag{1}$$

$$D_{gain} = 10\sqrt{1 - \left|\rho_{eij}\right|^2} \tag{2}$$

The value of ECC indicates the independent level among the radiation patterns of each MIMO elements. Likewise, the metric of DG is used to evaluate the improvement in signal quality when applying a diversity scheme. As shown in Fig 14, the DG values of the presented antenna within the operating band are approximately equal to the ideal value of 10 dB, while the figures for ECC are much lower than the acceptable value of 0.5, at around 0.001.

## Comparison

Table 2 compares the proposed prototype and the other related 2-port CP antennas using microstrip patch structure regarding their performance. The operating BW is defined by the overlap between -10 dB impedance and 3-dB AR BWs. In general, all the compared structures produce good isolation performance, with a minimum isolation of 20 dB. The proposed configuration in this paper is superior in edge-to-edge and center-to-center spacings compared to the reported ones. The reported antennas in [11, 27] offered narrow BWs. The wideband CP operation could be obtained in [30, 31, 33], but their drawbacks were low isolation and large inter-element spacings. Although the structure in [32] shows better operating BW and

**Table 2. Performance comparison among two-element CP antenna.**

| Ref. | Method | Edge spacing ($\lambda_0$) | Center spacing ($\lambda_0$) | Operating BW (%) | Iso. within BW (dB) |
|------|--------|---------------------------|------------------------------|------------------|---------------------|
| [11] | Grounded stubs, DGS | 0.06 | 0.47 | 2.2 | 20 |
| [27] | Grounded stubs, PE | 0.09 | 0.35 | 8.3 | 26 |
| [29] | MS superstrate, DGS | 0.14 | N/A | 0.75 | 24 |
| [30] | PE | 0.09 | 0.37 | 12.8 | 22 |
| [31] | MS | 0.19 | 0.44 | 13.7 | 20 |
| [32] | MS | 0.36 | N/A | 16.8 | 30 |
| [33] | Metallic post | 0.14 | 0.5 | 13.2 | 20 |
| Prop. | MS | 0.02 | 0.39 | 14.5 | 26 |

isolation, large spacing is also a critical disadvantage. To sum up, the proposed antenna offers various benefits, including wideband operation and high isolation while having small element spacing characteristics.

## Conclusion

The two-element MS-based antenna array with wideband operation and high isolation has been introduced and investigated in this paper. The proposed antenna exhibits dual-sense CP operation in the frequency range from 5.1 to 5.9 GHz. Besides, the isolation within the operating band is always better than 26 dB, while the realized gains are consistently higher than 4.2 dBi. Regarding the MIMO diversity performance, the calculated ECC and DG parameters demonstrate that the proposed configuration has an excellent diversity performance. With the outstanding performance as discussed above, the proposed antenna would be a potential candidate for applications in MIMO and full-duplex wireless communication systems.

## Author Contributions

**Conceptualization:** Hung Tran-Huy.

**Investigation:** Duc-Nguyen Tran-Viet, Hong Nguyen Tuan.

**Methodology:** Hong Nguyen Tuan, Dinh Nguyen Quoc.

**Supervision:** Hung Tran-Huy.

**Writing – original draft:** Duc-Nguyen Tran-Viet.

**Writing – review & editing:** Dinh Nguyen Quoc, Dat Nguyen Tien, Hung Tran-Huy.

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
