## [Decision Letter · Decision Letter 0]

2 Apr 2024

PONE-D-24-00283Metasurface-based dual-sense circularly polarized antenna for MIMO/full-duplex applicationsPLOS ONE

Dear Dr. Tran-Huy,

Thank you for submitting your manuscript to PLOS ONE. After careful consideration, we feel that it has merit but does not fully meet PLOS ONE’s publication criteria as it currently stands. Therefore, we invite you to submit a revised version of the manuscript that addresses the points raised during the review process.

We look forward to receiving your revised manuscript.

Kind regards,

Yuan-Fong Chou Chau

Academic Editor

PLOS ONE

Journal Requirements:

"This work was supported by the Vietnam Academy of Science and Technology (VAST) under grant number

TĐANQP.02/23–25"

4. We note that your Data Availability Statement is currently as follows: All relevant data are within the manuscript.

Reviewers' comments:

Reviewer's Responses to Questions

**Comments to the Author**

1. Is the manuscript technically sound, and do the data support the conclusions?

Reviewer #1: Yes

Reviewer #2: Yes

2. Has the statistical analysis been performed appropriately and rigorously? 

Reviewer #1: I Don't Know

Reviewer #2: Yes

3. Have the authors made all data underlying the findings in their manuscript fully available?

Reviewer #1: Yes

Reviewer #2: Yes

4. Is the manuscript presented in an intelligible fashion and written in standard English?

Reviewer #1: Yes

Reviewer #2: Yes

5. Review Comments to the Author

Reviewer #1: The authors have presented a two-element antenna array with dual-sense circular polarization, wideband operation, and high isolation characteristics. While the article is well-written and the discussions are engaging, several points need to be addressed before accepting this manuscript with minor revisions.

1. Describe the simulation method in the introduction section.

2. Elaborate on the novelty of the work in the introduction section. Clarify the unique properties of the proposed structures compared to existing literature.

3. The title of this manuscript relates to Metasurface-based polarized antenna, but there is no description of metamaterial and metasurface devices. To help the readers comprehend, address this issue and include other approaches of optical metasurface-based devices (Plasmonics, 2024, 19(1), 481–493, and https://doi.org/10.1007/s11468-024-02219-2) in the introduction section.

4. Fig. 2 illustrates the simulation result of the designed structure, but there is no information on the simulation methods in the text. Please clarify the simulation method and setting in more detail in the text.

5. Clarify the impact of structural parameters on device performance in the text.

6. Clarify in more detail the mechanism and the result discussion of Figs. 5, 7, 12, 13, and 14.

7. Ensure to check and correct any typos and grammatical errors throughout the manuscript.

Reviewer #2: In this paper, a two-element MS-based antenna array with wideband operation and high isolationhas been investigated. The paper is Well written and well presented. The contributions are novel and justified.

However, include state of the art work to further strengthen the literature, like https://doi.org/10.1371/journal.pone.0288793

https://doi.org/10.3390/sym15091641

https://doi.org/10.1155/2021/2286011

Recommended for publication in PLOS ONE.

6. PLOS authors have the option to publish the peer review history of their article (what does this mean?). If published, this will include your full peer review and any attached files.

Reviewer #1: No

Reviewer #2: No

---

## [Author Response · Author response to Decision Letter 0]

4 Apr 2024

Original Manuscript ID: PONE-D-24-00283

Original Article Title: “Metasurface-based dual-sense circularly polarized antenna for MIMO/full-duplex applications”

To: Reviewer

Re: Response to reviewer

Dear Reviewer,

We appreciate you for your precious time in reviewing our paper and providing valuable comments. It was your valuable and insightful comments that led to possible improvements in the current version. The authors have carefully considered the comments and tried our best to address every one of them.

We are uploading our point-by-point response to the comments, an updated manuscript with red highlighting indicating changes, and a manuscript without track changes.

Best regards,

 

Reviewer 1: The authors have presented a two-element antenna array with dual-sense circular polarization, wideband operation, and high isolation characteristics. While the article is well-written and the discussions are engaging, several points need to be addressed before accepting this manuscript with minor revisions.

Comment #1: Describe the simulation method in the introduction section.

Author response: Agreed.

Author action: The simulation tool is mentioned in Paragraph 4, Section Introduction of the revised manuscript.

Comment #2: Elaborate on the novelty of the work in the introduction section. Clarify the unique properties of the proposed structures compared to existing literature.

Author response: Agreed.

Author action: The novelty of the proposed work is briefly mentioned in Paragraph 4, Section “Introduction” of the revised manuscript.

Comment #3: The title of this manuscript relates to Metasurface-based polarized antenna, but there is no description of metamaterial and metasurface devices. To help the readers comprehend, address this issue and include other approaches of optical metasurface-based devices (Plasmonics, 2024, 19(1), 481–493, and https://doi.org/10.1007/s11468-024-02219-2) in the introduction section.

Author response: Agreed.

Author action: The suggested reference is included in the revised manuscript as ref [21, 22].

Comment #4: Fig. 2 illustrates the simulation result of the designed structure, but there is no information on the simulation methods in the text. Please clarify the simulation method and setting in more detail in the text.

Author response: Agreed.

Author action: Further discussion about the simulation method and setting is added to Paragraph 2, Section “Single-element design”. 

Comment #5: Clarify the impact of structural parameters on device performance in the text.

Author response: The authors do not fully understand the Reviewer’s comment. We are apologizing for this inconvenience. We are willing to address this properly with further explanation.

In fact, all the important structural parameters of the proposed design have been mentioned in Section “Optimization process”.

Comment #6: Clarify in more detail the mechanism and the result discussion of Figs. 5, 7, 12, 13, and 14.

Author response: Agreed.

Author action: The result discussion of Fig. 5 is highlighted in Paragraph 2, Section “Antenna operation characteristic”. The discussion for Fig. 7 is added to paragraph 1, Section “Matching optimization”. Figs. 12 and 13 show the comparison between simulations and measurements, the reason for the difference is highlighted in Paragraph 1, Section “Measured results”. The discussion for Fig. 14 is further added to Paragraph 1, Section “MIMO parameters”.

Comment #7: Ensure to check and correct any typos and grammatical errors throughout the manuscript.

Author response: Agreed.

Author action: The paper has been thoroughly checked.

Reviewer 2: In this paper, a two-element MS-based antenna array with wideband operation and high isolation has been investigated. The paper is Well written and well presented. The contributions are novel and justified. 

However, include state of the artwork to further strengthen the literature, like:

https://doi.org/10.1371/journal.pone.0288793

https://doi.org/10.3390/sym15091641

https://doi.org/10.1155/2021/2286011

Author response: Agreed.

Author action: The suggested references are cited in the revised manuscript as ref [1–3].

---

## [Editor Report · Decision Letter 1]

10 Apr 2024

Metasurface-based dual-sense circularly polarized antenna for MIMO/full-duplex applications

PONE-D-24-00283R1

Dear Dr. Tran-Huy,

We’re pleased to inform you that your manuscript has been judged scientifically suitable for publication and will be formally accepted for publication once it meets all outstanding technical requirements.

Kind regards,

Yuan-Fong Chou Chau

Academic Editor

PLOS ONE
---

## [Editor Report · Acceptance letter]

23 May 2024

PONE-D-24-00283R1 

PLOS ONE

Dear Dr. Tran-Huy, 

I'm pleased to inform you that your manuscript has been deemed suitable for publication in PLOS ONE. Congratulations! Your manuscript is now being handed over to our production team.

Kind regards, 

on behalf of

Dr. Yuan-Fong Chou Chau 

Academic Editor

PLOS ONE